# Dietary DHA Enhanced the Textural Firmness of Common Carp (*Cyprinus carpio* L.) Fed Plant-Derived Diets through Restraining FoxO1 Pathways

**DOI:** 10.3390/foods11223600

**Published:** 2022-11-11

**Authors:** Zijie He, Chao Xu, Fang Chen, Yunkun Lou, Guoxing Nie, Dizhi Xie

**Affiliations:** 1Guangdong Laboratory for Lingnan Modern Agriculture, College of Marine Sciences, South China Agricultural University, Guangzhou 510642, China; 2Laboratory of Aquatic Animal Nutrition and Diet, College of Fisheries, Henan Normal University, Xinxiang 453007, China

**Keywords:** DHA supplementation, common carp, textural firmness, MAPK/FoxO1pathway, PI3K/FoxO1 pathway

## Abstract

**Highlights:**

Dietary DHA of 0.42% was beneficial to the growth of common carp fed plant-derived diets;Dietary DHA ameliorated the muscular fatty acid nutrition, and textural firmness of common carp fed plant-derived diets; Myogenic regulatory factors expression was significantly elevated, while *mstna* expression was decreased by dietary DHA;Dietary DHA modification to the muscular textural firmness of common carp may be attributed to the inhibition of FoxO1 pathways

**Abstract:**

Omega-3 fatty acids have a positive effect on the muscle textural firmness of fish, while the intrinsic mechanism is poorly understood. To investigate the potential mechanism of textural modification caused by dietary docosahexaenoic acid ( DHA) in common carp (*Cyprinus carpio* L.), three plant-derived diets with varying DHA levels (0%, 0.5%, 1%, D1–D3) were prepared to feed juveniles (initial weight 15.27 ± 0.77 g) for 8 weeks, and the muscular texture, fibers density, and transcriptome were analyzed. The results showed that the growth performance, muscular DHA content, fibers density, and texture of the fish fed diets D2 and D3 were significantly ameliorated compared with the fish fed diet D1. The muscular transcriptome profiles indicated that the up-regulated genes of fish fed dietary DHA mainly in response to muscle proliferation, as well as the FoxO pathway, were significantly enriched in the D2 and D3 groups. Consistent with this, the Quantitative Real-Time PCR (qRT-PCR ) assays indicated that the expression of myogenic regulatory factors (*myog*, *myod*, *mrf4*, *mrf5*) was up-regulated in the high-DHA groups. Additionally, the expression of *foxo1* (inhibitor of myofiber development) mRNA was down-regulated, while its negative regulatory pathway (MAPK and PI3K) was activated in the D2 and D3 groups. The results suggested that the DHA supplementation is beneficial to modifying the muscular textural firmness of common carp fed plant-derived diets, which could be attributed to the inhibition of FoxO1 pathways.

## 1. Introduction

In recent years, with the increase of plant feedstuffs in the aquafeeds, the muscle quality, such as n-3 long-chain polyunsaturated fatty acids (n-3 LC-PUFA) contents and textural firmness, was impaired in cultured freshwater fish [1,2]. Although freshwater fish can biosynthesize n-3 LC-PUFA from α-linolenic acid (LNA, 18:3n-3) substrate, accumulating evidence now indicates that n-3 LC-PUFA-rich fish oil (FO) and microalgae supplementation play a positive influence on eicosapentaenoic acid (EPA, 20:5n3) and docosahexaenoic acid (DHA, 22:6n3) contents in the cultured fish [3]. Interestingly, the muscle textural firmness of fish was also modified by dietary n-3 LC-PUFA [4,5], while, the potential regulatory mechanism is unclear.

The muscle textural firmness of fish is closely associated with the myofiber number and size, which are determined by muscle fiber development [6]. The continual muscle development, characterized by hyperplasia (increases the muscle fibers’ number) and hypertrophy (increases the muscle fibers’ size), secures most of the life cycle in fish. Muscle development is regulated by multiple transcription factors, such as myogenic regulatory factors (Mrfs, including Myod, Myog, Mrf4 and Mrf 5) and myostatin (Mstn) [7]. It has been reported in mammals that the EPA treatments increased the mRNA expression of *Myod* and *Myog* genes in the skeletal muscle of rats [8]. Additionally, the muscle atrophy induced by palmitic acid can be modified by the protection of DHA [9]. Moreover, the intake of n-3 LC-PUFA (EPA and DHA) stimulated rates of skeletal muscle mass by increasing both fiber hypertrophy, and satellite cell activity directly, by altering proliferation and differentiation [10]. These mammal data provide a clue for studying the influences of n-3 LC-PUFA supplementation on the muscle textural firmness in fish.

Common carp (*Cyprinus carpio* L.) is characterized by high productivity, low feed inputs, and suitabability for multiple culture patterns, having a long history of being cultured in China [11]. In 2021, the production of cultured common carp reached to 2.83 million tons, which rankdc fourth among the fish species in China [12]. However, cultured common carp products have tended to decline in muscle texture, specifically causing muscle softening and poor mastication. Within the above contexts, this study was performed to explore the influences of DHA supplementation on the muscle texture of common carp fed plant-derived diets, and its potential regulatory mechanism.

## 2. Materials and Methods

### 2.1. Diets and Animal Management

Animals were treated according to the regulations of the South China Agricultural University Animal Protection and Use Committee (SCAU-AEC-2010-0416, approved on 17 January 2020). Using soybean meal, cottonseed meal, and rapeseed meal as the main protein sources, and soybean oil, DHA purified oil (contains 83.15% DHA) as the lipid sources, three diets (D1–D3) containing isonitrogenous (31%) and isolipidic contents (7%) were formulated. Diets D1–D3 were supplemented with DHA purified oil at 0% (control diets), 0.5%, and 1.0%, respectively. The preparation of diets was carried out, as we previously reported in detail [5]. The experimental diets were air-dried at room temperature and then stored in a refrigerator at −20 °C until used. The ingredient, proximate and fatty acid (FA) compositions of the three diets were shown in Table 1.

Common carp juveniles (body mass around 11 g,) were provided by local fisheries (Xinxiang, China). Before the rearing experiment, all fish were fed the control diets for 2-week to acclimatize them to the feeding conditions. Then, 360 healthy fish (initial weight 15.27 ± 0.77 g) were randomly assigned to 9 cylindrical aquaria with a diameter and depth of 1 m and 40 fish per aquaria. During the 8-week trial, fish were fed to satiation 3 times a day (8:30 a.m., 1:00 p.m., 5:30 p.m.), half of the water in the aquaria was changed daily (6:30 p.m.), the water temperature was kept at 25 ± 2 °C, and oxygen saturation was maintained by aeration. Meanwhile, the photoperiod was maintained at 12 h light/12 h dark.

### 2.2. Samples Collection and Growth Performance Evaluation

After the 8-week trial, all fish were fasted for 24 h and anesthetized with MS-222 (10 mg/L) before sampling. To analyze the growth performance, all fish in each tank were individually weighed. Blood, muscle, and viscera samples were randomly collected from nine fish per aquaria (27 fish per group). Particularly, the blood was sampled from the caudal vasculature and stood for 30 min at 4 °C before being centrifuged for 10 min at 4 °C, 3500 rpm, and then the serum was rapidly frozen and stored at −80 °C until analysis. Then, the body length, as well as the body, splanchnic, and liver qualities were measured for evaluating the morphological parameters. The left side muscle was isolated from fish, and rapidly preserved at −80 °C until the transcriptome and gene expression analysis. The other side muscle was sampled for the detection of flesh quality parameters, histological fiber, and ultrastructure. In addition, the approximate composition of whole fish was detected by randomly collecting another fish per aquaria.

Weight gain rate (WGR), feed conversion rate (FCR), and specific growth rate (SGR) were analyzed to evaluate the growth performance. The morphological parameter was evaluated by hepatosomatic index (HSI), viscerosomatic index (VSI), and condition factor (CF). These above parameters were assessed using standard formulations as previously reported [13].

### 2.3. Proximal and FA Composition

The approximate components of the whole fish and diets were measured through the assay method described in our previous study [14]. Briefly, moisture was assayed by drying samples to a constant weight at 105 °C. Ash content was measured by incinerating samples in a muffle furnace at 550 °C for 6 h. Crude protein was evaluated by the Kjeldahl method, and the crude protein content was calculated as % N × 6.25. Crude lipid content was assayed by the Soxhlet method. Each sample was analyzed 3 times.

According to the common method, as previously reported [13], the lipid of diets and muscle was separated through a mixture of chloroform and methanol with a volume ratio of 2:1., and then transesterification was performed with boron trifluoride etherate to prepare fatty acid methyl esters (FAME). Finally, the FAME was dissolved in n-hexane and separated by gas chromatography (Agilent Technologies Inc., Palo Alto, CA, USA), as previously described [13]. Briefly, FAME samples were applied using an on-column (HP-88, 100 m × 0.25 mm × 0.2 μm, Agilent Technologies Inc., Palo Alto, CA, USA) injection, and the oven temperature was programmed to go from 80 to 250 °C, at a rate of 40 °C per minute. The fatty acid composition was identified according to the commercial FAME standards (Sigma-Aldrich, Saint Louis, MO, USA), and calculated by the area normalization method.

### 2.4. Biochemical Analysis

The serum samples were kept at a low temperature during testing. The serum biochemical indices, including total protein (TP), albumin (ALB), glucose (GLU), total cholesterol (CHO), high-density lipoprotein (HDL), low-density lipoprotein (LDL), triglycerides (TG), free fatty acid (NEFA), alkaline phosphatase (AKP), acidic phosphatase (ACP), superoxide gasification enzyme (SOD), catalase (CAT), malondialdehyde (MDA) and glutathione peroxidase (GSH-Px), were measured via the use of an enzyme-linked immunosorbent assay (ELISA) method. All these indicators were analyzed according to the detection scheme of mercantile reagent kits (Nanjing Jiancheng Biochemical Corporation, Nanjing, China).

### 2.5. Muscle Nutritional Indices and Texture Characteristics Test

Fish lipid quality/flesh lipid quality (FLQ), polyunsaturated fatty acid/saturated fatty acid ratio (PUFA/SFA), n-6/n-3 PUFA, index of thrombogenicity (IT), and hypocholesterolemic/hypercholesterolemic ratio (HH) are the indices for evaluating the muscle nutrition of fatty acids. The values of these above indices were calculated based on the data of FA contents in muscle using standard formulations, as previously reported [13].

The muscular textural properties, like tenderness, hardness, resilience, chewiness, etc., were evaluated by a Texture Analyzer (Universal TA, Shanghai turnkey pull, Shanghai, China), with the specific parameters set according to our previous study [15]. The muscular edibleness characteristics, including water-holding capacity (WHC), cooking percentage (COP), and potential of hydrogen (pH), were measured by the method developed in research by Ma et al. [15].

### 2.6. Histological and Ultrastructural Observation

For histological observation, the muscle samples were cut into 8-micron slices (8 μm) by a cryostat microtome, and cold-fixed in 10% buffered formalin for 10 min. Then, the sections were stained with modified oil-red O (G1261, Solarbio, Corp., Beijing, China), followed by light microscopy observations [16]. For ultrastructural observation, the muscle samples were soaked in glutaraldehyde solution and rinsed with phosphate-buffered saline. Then, the samples were fixed in an osmium tetroxide aqueous solution and washed with a phosphate buffer, and then dehydrated with different concentrations of ethanol and inlaid in epikote for the electron microscope observation. 

### 2.7. Transcriptome Analysis

The total RNA of the white muscle was isolated from three fish per tank and pooled into a mixture sample, and three samples per dietary group were prepared for transcriptome analysis. RNA Sequencing Library Preparation and Deep Sequencing were performed by a Biomedical Technology Company (Shanghai Meiji Biomedical Technology Co., Ltd., Shanghai, China). The statistical analysis of transcript levels was assessed by the DESeq R software package. Based on the *p* values, the method of Benjamini and Hochberg was adopted to calculate the false discovery rate (FDR) and the differently expressed genes were identified when FDR ≤ 0.05, |logFC| ≥ 1. These transcripts were considered to be down- or up-regulated when the log2 fold change was ≤1 or ≥1, respectively. In addition, with the use of Gene Ontology (GO) and the Kyoto Encyclopedia of Genes and Genomes (KEGG) pathway enrichment analysis, the function of differently expressed genes (DEGs) was explored.

### 2.8. RNA Extraction and Quantitative Real-Time PCR (qRT-PCR)

To verify the accuracy of RNA-Seq, data were verified by qRT-PCR of RNA samples of the transcriptome. The expression of 13 differently expressed genes, including *myog*, *myod*, *mrf4*, *mrf5*, *mstna*, *mstnb*, *pi3k*, *pdk*, *sgk*, *ras*, *raf*, *mapk*, and *foxo1*, related to myofiber development and FoxO signaling pathways were detected by qPCR. RNA extraction, cDNA synthesis, and qPCR assays were executed as described previously [13]. Briefly, 1 μg total RNA was extracted from the muscle by RNAiso Plus, and reverse-transcribed into cDNA through an RT-PCR kit (Takara Biomedical Technology Co., Ltd., Dalian, China) in a system containing random primers. The qPCR was performed in a Lightcycler 480 system (Roche, Basel, Switzerland). The initial DNA was denatured for 3 min at 94 °C, followed by 45 cycles at 95 °C for 15 s, annealed for 10 s at 60 °C, and extended for 10 s at 72 °C. The qPCR data were calculated by the 2^−ΔΔCT^ method, with the 18S rRNA as the internal control [17]. The primer pairs for qPCR were designed (Appendix A). The reaction was repeated 3 times for each sample.

### 2.9. Statistical Analysis

All data were subjected to one-way ANOVA and Duncan’s test by SPSS 22.0 (SPSS Inc., Chicago, IL, USA), and expressed as mean ± standard error of means (SEM). When the difference between groups was significant, Tukey’s multiple comparisons test was executed and considered significant with *p* < 0.05.

## 3. Results

### 3.1. Growth and Proximate Composition of Fish

As shown in Table 2, the WGR and SGR in the D2 treatment were remarkably higher than in the D1 treatment (*p* < 0.05), while no obvious differences were measured in FCR, HIS, VSI, and CF among the three groups (*p* > 0.05). Similarly, there were no significant differences in the proximate composition of the whole body among the three treatments (*p* > 0.05).

### 3.2. Biochemical Analysis

Serum TG, LDL, HDL, and GLU levels, as well as AKP and ACP enzymatic activities, showed significant differences among the four dietary DHA treatments (*p* < 0.05) (Table 3). Compared to the D1 group, lower serum TG and LDL levels, and higher serum HDL and GLU levels were displayed in the D2 and D3 treatments. Particularly, the serum AKP and ACP activities in D2 and D3 groups showed significantly high than those of the D1 group (*p* < 0.05), while dietary DHA had no significant influence on the serum TP, NEFA, and MDA contents, as well as the serum SOD, CAT, and GSH-Px activities (*p* > 0.05).

### 3.3. FA Compositions and Nutritional Indices of Muscle

The FA profiles of muscle were shown in Table 4. The n-3 PUFA contents increased in parallel with the increase of dietary DHA levels, whereas n-6 PUFA and MUFA showed an opposite trend (*p* < 0.05). Particularly, higher levels of muscular ALA, EPA, and DHA were observed in the fish treated with diets D2 and D3 (*p* < 0.05), and the n-3 PUFA contents were enhanced 1.37–1.99-fold compared with fish fed the D1 diet. While significantly low muscle LA and n-6 PUFA contents were detected in the D2 and D3 groups (*p* < 0.05). In the aspect of the nutritional indices of FA in muscle (Table 5), the nutritional indices of muscle FLQ were significantly more improved in the D2 and D3 groups than those of D1 dietary treatment, and the n-6/n-3 PUFA ratio and IT of muscle in the D2 and D3 groups decreased by 35.16–56.32% and 15.80–31.58% compared with the D1 group (*p* < 0.05).

### 3.4. Proximate Composition, Edible Quality, and Textural Properties of Muscle

The muscular proximate composition, edible quality, and textural properties were shown in Table 4. The muscle moisture, crude protein and lipid, as well as pH, and COP shared comparable values when fish were treated with three diets (*p* > 0.05), while the muscle WHC was significantly ameliorated by dietary DHA (*p* < 0.05). Additionally, the gumminess of textural properties in muscle was elevated in the D2 and D3 dietary treatments compared with the D1 dietary treatment (*p* < 0.05) (Table 5). Although there was no statistical difference in muscular hardness among the three groups, the quantitative value of those indices in the D2 and D3 dietary treatments was more substantial than that of the D1 dietary treatment. However, no significant difference was observed in muscular tenderness, adhesiveness, springiness, and chewiness among the three dietary treatments (*p* > 0.05).

### 3.5. Histological Fiber and Ultrastructure Analysis

The histological characteristics of the muscle, including myofiber diameter and density, myofiber sarcomere length, and fillet lipid distribution, were shown in Figure 1A–E. The morphological structure of myofiber is irregular polygons, the myofiber diameters showed a significant decrease with the increase of dietary DHA among the three groups (*p* < 0.05) (Figure 1A–D), whereas the myofiber densities showed increasing trends in the dietary DHA treatments (*p* < 0.05) (Figure 1A–C,E). The ultrastructure of muscle myofiber was shown in Figure 1F–I. The muscular sarcomere lengths of fish treated with diets D2 and diets D3 were significantly high compared with those treated with the D1 dietary treatment (*p* < 0.05).

### 3.6. GO and KEGG Enrichment Analysis of DEGs

To explore the potential molecular mechanisms of changes in the myofiber caused by dietary DHA, the muscle transcriptome was analyzed. As shown in Appendix A, a total of 1257 DEGs (540 up-regulated and 717 down-regulated genes), 620 DEGs (211 up-regulated and 409 down-regulated), and 182 DEGs (110 up-regulated and 72 down-regulated) were measured from D1 and D2; D1 and D3; D2 and D3, respectively.

The biological functions of DEGs were identified by the gene ontology (GO) enrichment analysis. As shown in Appendix A, the 20 most enriched GO terms, including four cellular components, eight biological processes, and eight molecular functions, were observed in the DEGs between the D1 and D2 groups. Additionally, six cellular components, eight biological processes, and six molecular function terms were significantly enriched between D1 and D3 groups (Appendix A). Additionally, 16 enriched GO pathways associated with muscle development were observed among the three groups (Appendix A). A cluster heat map was built from 55 DEGs related to myofiber development, which shows a distinct expression pattern among the different groups (Figure 2 and Appendix A). Specifically, 44 DEGs, including the myogenic regulatory factors (*myod*, *mrf5*, *myog*, *mrf4*), *myosin*, and *mapk*, were up-regulated in the muscle of D2 and D3 dietary treatments (*p* < 0.05), while 11 DEGs, including the *mstn* and RNA-binding protein 24, were down-regulated in the fish treated with D2 and D3 diets.

The top 15 enriched KEGG pathways between D1 versus D2 (Appendix A), D1 versus D3 (Appendix A) were the ribosome, FoxO signaling pathways, oxidative phosphorylation, Parkinson’s disease, non-alcoholic fatty liver disease (NAFLD), circadian, rhythm, and microRNAs in cancer. Interestingly, the FoxO signaling pathways were both presented in D1 versus D2, and D1 versus D3. As shown in the FoxO signaling pathways (Appendix A), the FoxO was mainly regulated by 3 pathways, which include two negative regulatory pathways (MAPK and PI3K) and one positive regulatory pathway (AMPK). Specifically, compared to the D1 group, the MAPK (*egfr*, *grb2*, *sos*, *ras*, *raf*, *mek*, and *erk*) and PI3K (*pi3k* and *sgk*) pathway components were up-regulated, and the AMPK pathway was down-regulated with the D2 and D3 dietary treatments. Additionally, the down-regulation of *foxo* was detected in the D3 treatment compared with that in the D1 treatment (*p* < 0.05).

In addition, 43 DEGs related to the FoxO signaling pathway also showed a distinct expression pattern among the different groups (Figure 3 and Appendix A). Specifically, 29 DEGs, including the MAPK and PI3K pathway components, and insulin receptors, were up-regulated in the muscle of D2 and D3 groups (*p* < 0.05), while, compared to the D1 group, 14 DEGs, including the FoxO pathways, were inhibited in the D2 and D3 groups (*p* < 0.05).

### 3.7. RNA-Seq Results Were Verified by qRT-PCR 

To verify the reliability of the transcriptome profile, 13 key genes (*myog*, *myod*, *mrf4*, *mrf5*, *mstna*, *mstnb*, *pi3k*, *pdk*, *sgk*, *raf*, *mek*, *erk*, and *foxo1*) related to myofiber development and FoxO signaling were selected for qPCR confirmation. The consistency of RNA-Seq and qRT-PCR results data (Figure 4) indicated the reliability of RNA-Seq assembly. Compared to the D1 group (without DHA), the mRNA expression level of myogenic regulatory factors (*myod*, *mrf5*, *myog*, *mrf4*), MAPK and PI3K pathway components (*pi3k*, *pdk*, *sgk*, *raf*, *mek*, and *erk*) in the muscle of D2 and D3 dietary treatments(high-DHA contents) were higher (*p* < 0.05), while the *foxo1* and *mstna* mRNA levels in the muscle of D2 and D3 dietary treatments were lower (*p* < 0.05), and the *mstnb* mRNA displayed similar levels to those found among the three treatment groups.

## 4. Discussion

Like other freshwater fish species, common carp can synthesize LC-PUFA, using C18 PUFA as a substrate (like LA and ALA) through a cascade of fatty acid desaturase (Fads) and elongase of very-long-chain fatty acids (Elovls). Thus, its essential fatty acids (EFA) are LA and ALA [3]. While increasing evidence showed that the supplementation of dietary n-3 LC-PUFA is beneficial to the growth of freshwater fish [18,19], in this study, the WGR and SGR of common carp treated with diets D2 were significantly higher than those of D1 dietary treatment. The study of grass carp (*Ctenopharyngodon idellus*) found that the WGR, SGR, feed efficiency (FE), and protein efficiency were improved when the dietary LC-PUFAs content increased from 0% to 0.52% [20]. While it was nonlinear in the growth of fish treated and the dietary DHA levels, the D3 group showed comparable growth with the D1 group. Similar results have been reported on blunt snout bream (*Megalobrama amblycephala*) [4], gibel carp (*Carassius auratus gibelio*) [21], Nile tilapia (*Oreochromis niloticus*) [22], and largemouth bass (*Micropterus salmoides*) [23]. This indicates that the supplementation of 0.42% DHA (DHA purified oil containing 83.15% DHA) in plant-derived diets is beneficial to improve the growth of common carp.

In addition to ameliorating the growth performance, dietary n-3 LC-PUFA also had a positive influence on the lipid metabolism of fish [5,24]. Recent studies demonstrated that there was an appropriate dietary n-3 LC-PUFA-level ameliorated serum TG, LDL, and HDL levels in the mirror carp (*Cyprinus carpio var. specularis*), golden pompano (*Trachinotus ovatus*), California Yellowtail (*Seriola dorsalis*), and hybrid grouper (*♀ Epinephelus fuscoguttatus × ♂ Epinephelus lanceolatu*) [5,24,25,26]. Similarly, the serum lipid metabolite concentrations of common carp, snakehead (*Channa argus*), and pirarucu (*Arapaima gigas)* were attenuated by the plant-derived diets [27,28]. In the present study, high serum HDL levels, and low serum LDL and TG levels were observed in the common carp fed dietary DHA (D2 and D3). The results indicated that the n-3 LC-PUFAs supplementation is beneficial to the lipid metabolism and healthy growth of fish [20,29].

Compared to the EFA requirement of growth and health, maintaining fatty acid nutrition needs more EFA in cultured fish [3]. In our study, high growth was demonstrated in the D2 group, while there were consistent changes in the muscular n-3 LC-PUFA contents and dietary DHA content. In the study of Nile tilapia, the deposition level of muscle DHA was in line with the dietary *Schizochytrium sp.* (rich in DHA) level [30]. Additionally, with the increase of muscular n-3 LC-PUFA content, the nutritional indices of FAs were significantly modified in the common carp treated with diets D2 and D3, such as n-6/n-3 PUFA, FLQ, and IT. Similarly, the n-3/n-6 PUFA and FLQ of filet of silvery-black porgy (*Sparidentex hasta*) were significantly improved in the FO dietary treatment [31]. Álvarez et al. [32] indicated that gilthead seabream (*Sparus aurata*) fed with the plant oil (VO) diets showed lower n-3/n-6 ratio, FLQ, and AI than those given FO dietary treatment. These results indicate that the muscular FA nutritional value of fish can be significantly elevated by the supplementation of n-3 LC-PUFA.

In addition to the muscle FA nutrition, the myofiber structure and muscle texture characteristics were also ameliorated by dietary DHA. In this study, the myofiber density and sarcomere length of common carp in D2 and D3 treatments were significantly increased, while the fiber diameter was significantly decreased. These results are in line with the finding of the study of blunt snout bream, which indicated that high dietary DHA levels (0.8%) enhance the fiber density and sarcomere lengths [4]. In the study of Senegalese sole (*Solea senegalensis*), the fiber density in the VO dietary treatment was decreased, while the fast-twitch fiber diameter and dorsal muscle cross-sectional area were increased, compared with fish of FO dietary treatment (high n-3 LC-PUFA contents) [33]. These results indicate that dietary n-3 LC-PUFA is beneficial to stimulating muscle hyperplastic growth (high fibers density, and low myofibers diameter) [33].

Previous studies found that fiber density is correlated with muscle hardness, springiness, chewiness, and gumminess [4,34]. In blunt snout bream treated with the high DHA, the myofiber density was significantly increased; accordingly, the hardness, chewiness, and gumminess of flesh were also improved [4]. In the present study, the muscle textural firmness, like the change of fiber density, in the D2 and D3 dietary treatments (containing 0.5–1.0% DHA purified oil) are significantly higher than the D1 dietary treatment (soybean oil). Compared to the wild sea bass (*Dicentrarchus labrax* L.), the cultured fish species has low muscle DHA levels, and its myofiber density and textural firmness are significantly inferior to those of wild fish species [34]. According to these above results, supplementation of DHA modified the textural firmness properties by stimulating the muscle hyperplastic growth of the farmed common carp fed plant-derived diets.

Myoblast proliferation and differentiation contribute to the hyperplastic growth and hypertrophic growth of fish, which is regulated by a series of positive factors (Myod, Mrf5, Mrf4, Myog) and negative factors (Mstn) [6,7]. Myod and Myf5 play a role in activating genes involved in cell cycle progression and leading to myoblast proliferation, while Myog and Mrf4 mediate differentiation and induce cell cycle exiting. Then, these above myogenic regulatory factors synergize to drive muscle differentiation [35]. An increasing number of studies indicated that n-3 LC-PUFA involves the regulation of myogenesis at the transcriptional level [35]. Wang et al. [4] found that dietary DHA induces the muscular *myog* and *mrf4* expression and increases the myofiber development of blunt snout bream. In our study, the expressions of *myod*, *myf5*, *mrf4*, and *myog* were up-regulated in the muscle of DHA dietary treatments (D2 and D3 groups), while the negative factor *mstna* was decreased. In mammals, the effects of DHA treatment on the differentiated C2C12 myoblasts also showed an up-regulation of *mrf4*, *myog*, and *myod* mRNA expression levels, and an increase in myogenic differentiation [35]. The results indicated that dietary DHA has the potential to regulate the complex process of muscle proliferation and differentiation due to its ability to regulate the expression of myogenic factors.

Besides myogenic factors, the proliferation and differentiation of myofibers can be regulated by the upstream transcription factors, such as FoxO1 [36]. A series of mammalian studies revealed that FoxO1 is an inhibitor of myofiber development, which repressed the expression of *myod* and its promoter activity, and increased Mstn expression [36]. In this study, the abundance of *foxo1* was decreased in the muscle of common carp fed diets supplemented with DHA. Accordingly, high myogenic factor expression and low *mstna* expression were detected in dietary DHA groups (D2-D3). In line with these results, the diets supplemented with DHA had a decrease in FoxO1 expression in the adipose and liver of weaned cross-bred pigs, and DHA treatments also reduced the activity of FoxO1 promoter in humans [37]. Liu et al. [38] reported that fish oil decreased *foxo1* mRNA abundance, and increased muscle protein mass. These studies suggested that dietary DHA negatively mediates FoxO1 to influence the expression of myogenic regulatory factors.

Moreover, FoxO activation is also regulated by a variety of upstream signaling pathways, which includes the inactivation of mitogen-activated protein kinase (MAPK) and phosphatidylinositol 3-kinase (PI3K) signaling pathways [36,39,40]. In the present study, signaling pathway activities associated with PI3K (*pi3k*, *pdk*, and *sgk*) and MAPK (*raf*, *mek*, and *erk*) were evaluated in the common carp fed high-DHA diets (D2 and D3). These discoveries are consistent with a previous mammal study that reported that n-3 LC-PUFA induces airway smooth muscle cell proliferation through PI3K and MAPK pathways [41]. The results suggested that the dietary DHA induces the decrease of *foxo1* expression level by activating the PI3K and MAPK pathways, which supplementation of DHA modified the muscle hyperplastic growth and textural firmness of the common carp fed plant-derived diets. 

## 5. Conclusions

In conclusion, given the results of growth performance, muscular fatty acid nutrition, and textural properties, the present study demonstrated that dietary DHA can modify the growth and textural firmness of common carp fed plant-derived diets. Moreover, the dietary DHA is beneficial for the textural firmness of common carp by inducing muscle hyperplastic growth, which could be attributed to the activation of PI3K and MAPK pathways, and the inactivation of the inhibitor of myofiber development—FoxO1. These results provide new insights for understanding the potential mechanisms of DHA supplementation to improve the textural firmness of common carp fed plant-derived diets.

## Figures and Tables

**Figure 1 foods-11-03600-f001:**
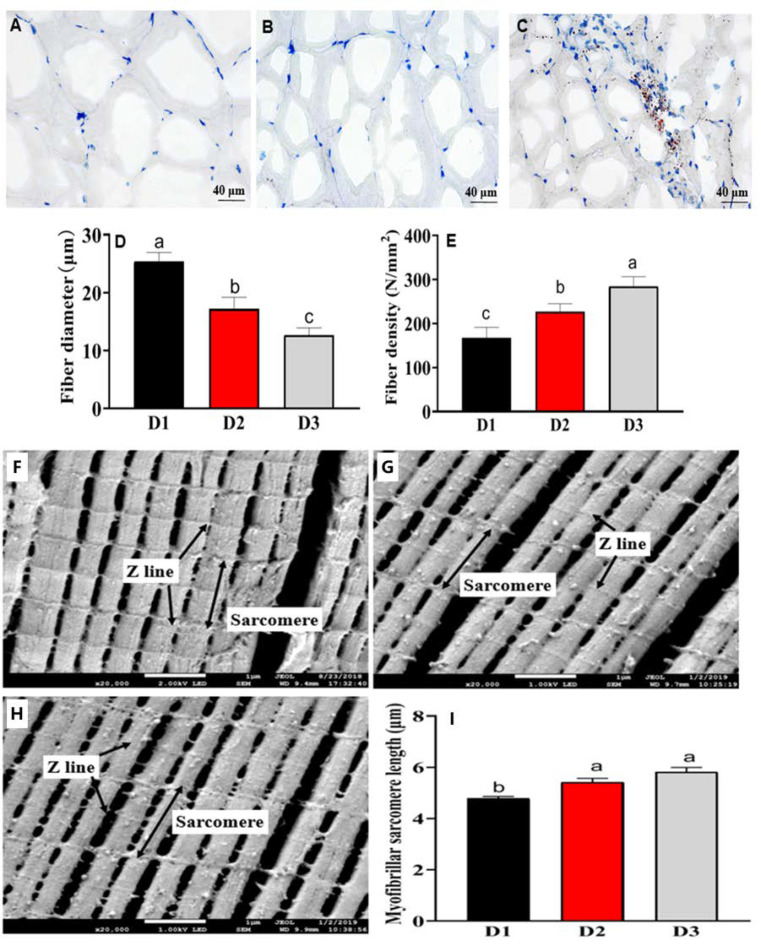
Transverse section microstructure and transmission electron microscope of muscle in common carp fed experimental diets. Transverse section microstructure (**A**–**C**): oil-red O staining, ×400, scale bar 40 μm. (**A**) D1 diet treatment group; (**B**) D2 diet treatment group; (**C**) D3 diet treatment group; (**D**) myofiber diameter of common carp; (**E**) myofiber density of common carp. Transmission electron microscope: (**F**–**H**): photomicrographs (×20,000) and scale bar (1 μm). The lines with bidirectional arrows indicated the sarcomere lengths. The lines with an unfilled arrow indicated the Z line. (**F**) D1 diet treatment group; (**G**) D2 diet treatment group; (**H**) D3 diet treatment group; (**I**) myofibrillar sarcomere length of common carp. Values are expressed as means ± SEM (*n* = 3). Bars among each group with different letters indicated significant differences (*p* < 0.05).

**Figure 2 foods-11-03600-f002:**
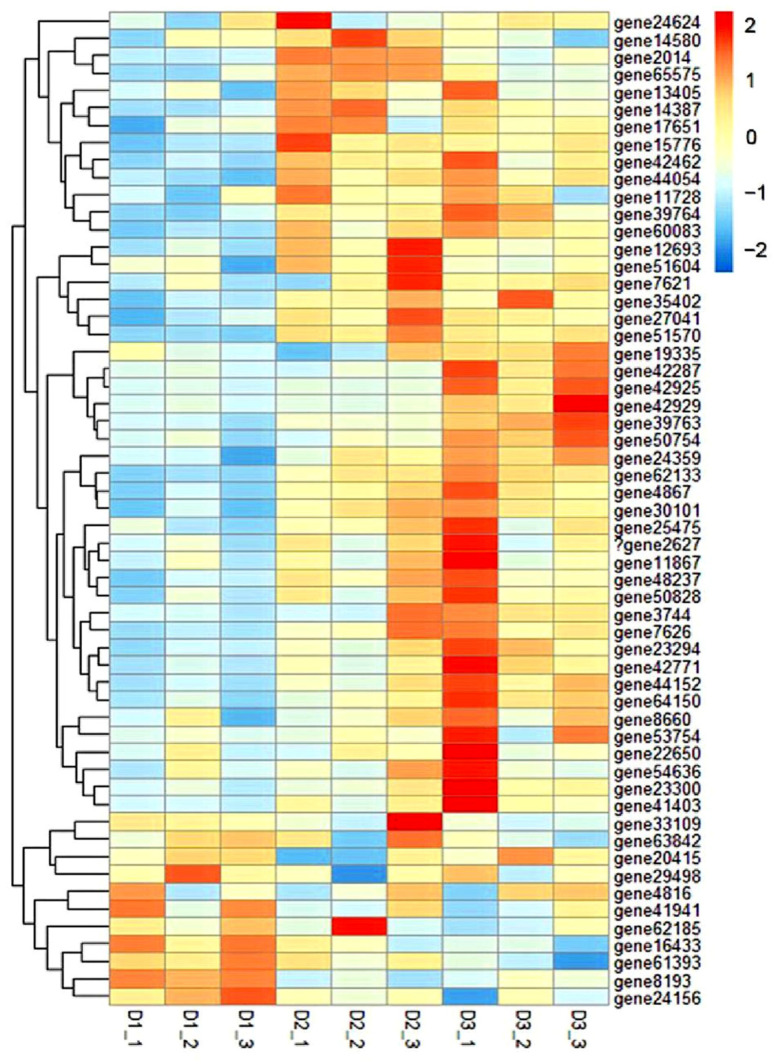
Clustered heat map of the expression of genes related to myofiber development in common carp fed with diets D1–D3. The distance algorithm (inter-sample Spearman and inter-gene Pearson correlation coefficients) was used for cluster analysis. Transcript enrichment is indicated in the heat map from low (blue) to high (red), as shown in the colored groups to the right of the heat map.

**Figure 3 foods-11-03600-f003:**
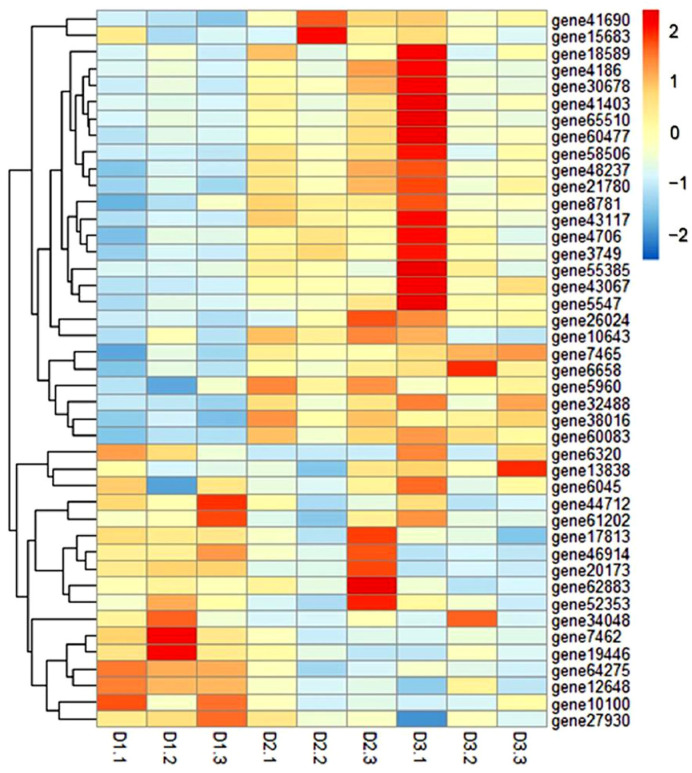
Clustered heat map of the expression of genes related to FoxO signaling pathways in common carp fed with diets D1–D3. The distance algorithm (inter-sample Spearman and inter-gene Pearson coefficients) was used for cluster analysis. Transcript enrichment is indicated in the heat map from low (blue) to high (red), as shown in the colored groups to the right of the heat map.

**Figure 4 foods-11-03600-f004:**
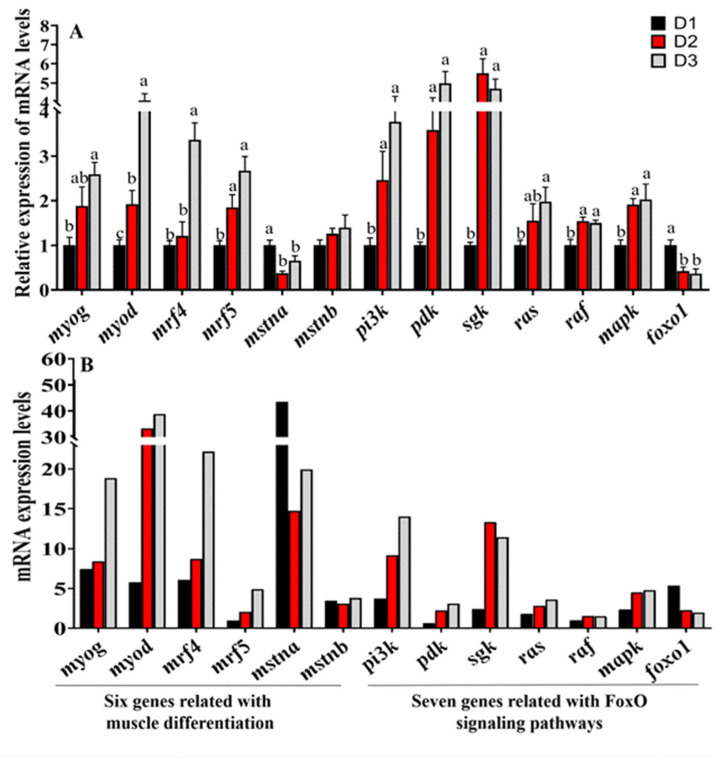
The gene expression profile was verified by qRT-PCR. The relative expression trends between RNA-seq and qRT-PCR of the genes related to muscle differentiation and FoxO signaling pathways were completely consistent. (**A**) is qRT-PCR results; (**B**) is RNA-Seq results. Values are expressed as means ± SEM (n = 3). Bars among each gene with different letters indicate significant differences. (*p* < 0.05).

**Table 1 foods-11-03600-t001:** Ingredients, proximate composition, and FA composition of diets.

Ingredients (% Dry Matter)	Experimental Diets		
	D1	D2	D3
Soybean meal	40	40	40
Rapeseed meal	25	25	25
Cottonseed meal	10	10	10
Soybean oil	6	5.5	5
DHA purified oil ^a^	0	0.5	1
Proxim compound ^b^	1.0	1.0	1.0
Others ^c^	18.1		
**Proximate composition (% dry matter)**			
Dry matter	90.48	90.54	90.78
Crude protein	31.43	32.23	31.45
Crude lipid	6.88	6.94	7.03
Ash	6.77	6.86	6.15
**Main FA composition (% total FAs)**			
14:0	0.58	0.59	0.57
16:0	6.38	5.9	7.35
18:0	2.9	2.76	2.66
20:0	0.46	0.6	0.62
16:1	0.65	0.72	0.56
18:1n9	20.64	18.57	18.29
18:2n6	55.49	49.61	41.64
18:3n3	6.12	6.72	7.09
20:4n6	/	0.47	0.7
20:5n3	/	0.89	1.21
20:6n3	/	5.98	9.3

^a.^ DHA-enriched oil contains 8.96% SFA, 4.89% EPA, and 83.15% DHA, Xi’an Sheng he Biological Technology Co., Ltd., Xi’an, China. ^b.^ Consist of vitamin mixture (0.5%) and mineral compound (0.5%). The mixed vitamins in per kg: A, 1 × 10^6^ IU; D3, 3 × 10^5^ IU; E, 5000 IU; K3, 1040 mg; B1, 1500 mg; B2, 2400 mg; B6, 1200 mg; B12, 5 mg; nicotinic acid, 8000 mg; D-calcium pantothenate, 3200 mg; folic acid, 400 mg; biotin, 10 mg; inositol, 12,000 mg; C-monophopholipid, 16,000 mg. The mixed minerals per kg: iron, 10 g; zinc, 3.2 g; manganese, 3 g; cobalt, 52 mg; iodine, 65 mg; selenium, 15 mg. Chengyi Co., Ltd., Guangzhou, China. ^c.^ Others include wheat flour 10%, rice bran 6.7%, choline chloride 0.2%, dicalcium phosphate 1%, L-lysine 0.1%, DL-methionine 0.1%./indicated no detectable.

**Table 2 foods-11-03600-t002:** Growth, feed conversion rate, and proximate composition of common carp with different dietary treatments after 8 weeks.

	Experimental Diets		
	D1	D2	D3
IBW (g)	15.26 ± 0.06	15.21 ± 0.04	15.34 ± 0.05
FBW (g)	31.04 ± 1.27	32.92 ± 1.64	31.83 ± 1.42
WGR (%)	103.37 ± 4.16 ^b^	115.06 ± 3.88 ^a^	107.52 ± 4.09 ^a,b^
SRG (%)	1.27 ± 0.03 ^b^	1.37 ± 0.03 ^a^	1.30 ± 0.03 ^a,b^
FCR	1.38 ± 0.03	1.33 ± 0.02	1.29 ± 0.03
Survival (%)	100	100	100
HSI	5.16 ± 0.27	5.21 ± 0.26	5.18 ± 0.24
VSI	1.04 ± 0.06	1.08 ± 0.04	1.07 ± 0.12
CF (g/cm^3^)	2.57 ± 0.08	2.73 ± 0.16	2.66 ± 0.12
Proximate composition (% wet matter)			
Moisture	69.40 ± 0.53	70.25 ± 0.74	68.02 ± 0.72
Crude protein	17.31 ± 0.03	17.81 ± 0.34	17.53 ± 0.28
Crude lipid	8.17 ± 0.37	7.94 ± 0.44	7.50 ± 0.61
Ash	3.05 ± 0.04	3.56 ± 0.13	3.10 ± 0.09

Values are mean ± SEM (*n* = 3). It shows significant differences when the superscript letters of the values in the same line are completely different (*p* < 0.05). IBW: initial body weight; FBW: final body weight; WGR: weight gain rate; SGR: specific growth rate; FCR: feed conversion ratio; HIS: hepatosomatic index; VSI: viscerosomatic weight; CF: condition factor.

**Table 3 foods-11-03600-t003:** Serum biochemical parameters of common carp with different dietary treatments after 8 weeks.

Index	Experimental Diets		
	D1	D2	D3
TP (mmol/L)	23.27 ± 0.99	22.67 ± 1.49	22.07 ± 0.55
GLU (mmol/L)	4.74 ± 0.87 ^b^	5.17 ± 0.60 ^a,b^	6.28 ± 0.87 ^a^
TG (mmol/L)	1.05 ± 0.05 ^a^	0.72 ±0.08 ^b^	0.81 ± 0.05 ^b^
NEFA (mmol/L)	1.92 ± 0.36	2.05 ± 0.46	2.04 ± 0.44
CHO (mmol/L)	2.50 ± 0.13 ^a^	1.85 ± 0.17 ^b^	2.55 ± 0.19 ^a^
LDL (mmol/L)	1.00 ± 0.09 ^a^	0.65 ± 0.11 ^b^	0.82 ± 0.08 ^b^
HDL (mmol/L)	1.27 ± 0.05 ^b^	1.51 ±0.07 ^a,b^	1.90 ± 0.41 ^a^
ALB (mmol/L)	10.87 ± 0.75 ^a^	8.78 ±0.71 ^b^	12.58 ± 0.65 ^a^
ACP (U/mL)	5.24 ± 0.35 ^b^	7.59 ± 0.10 ^a^	7.68 ± 0.60 ^a^
AKP (U/mL)	5.97 ± 0.40 ^b^	7.01 ± 0.64 ^a^	7.86 ± 0.86 ^a^
SOD (U/mL)	136.72 ± 1.74	137.83 ±2.96	140.17 ± 3.80
GSH-Px (U/mL)	542.73 ± 33.78	492.45 ± 44.74	602.12 ± 15.28
CAT (U/mL)	94.68 ± 2.57	104.33 ± 3.35	107.65 ± 1.77
MDA (nmol/mL)	0.91 ± 0.04	0.72 ± 0.14	0.84 ± 0.25

Values are mean ± SEM (*n* = 3). It shows significant differences when the superscript letters of the values in the same line are completely different (*p* < 0.05). TP: total protein; GLU: glucose; TG: triacylglycerol; NEFA: free fatty acid; CHO: total cholesterol; HDL: high-density lipoprotein; LDL: low-density lipoprotein; ALB: albumin; ACP: acid phosphatase; AKP: alkaline phosphatase; GSH-Px: glutathione peroxidase; SOD: superoxide dismutase; CAT: Catalase; MDA: malondialdehyde.

**Table 4 foods-11-03600-t004:** FA composition of muscle in common carp with different dietary treatments after 8 weeks.

Index	Experimental Diets		
	D1	D2	D3
**Main FAs (% total FAs)**			
12:0	/	/	/
14:0	0.44 ± 0.02	0.48 ± 0.03	0.42 ± 0.02
16:0	16.48 ± 1.86	14.90 ± 1.47	15.30 ± 1.34
18:0	5.01 ± 0.36 ^a^	3.70 ± 0.11 ^b^	2.18 ± 0.46 ^b^
20:0	0.45 ± 0.04 ^a^	0.26 ± 0.01 ^b^	0.23 ± 0.02 ^b^
SFA	23.08 ± 2.06	21.04 ± 1.84	21.15 ± 1.76
16:1	6.80 ± 0.91	6.82 ± 0.42	6.73 ± 0.44
18:1	32.04 ± 1.83 ^a^	29.13 ± 1.12 ^a,b^	27.50 ± 1.06 ^b^
MUFA	40.02 ± 3.83 ^a^	37.12 ± 2.82 ^a,b^	36.45 ± 1.88 ^b^
18:2n-6	34.04 ± 3.58 ^a^	30.66 ± 0.99 ^a,b^	28.09 ± 3.32 ^b^
18:3n-6	1.72 ± 0.02 ^a^	1.33 ± 0.03 ^b^	1.26 ± 0.59 ^b^
20:4n-6	0.73 ± 0.06 ^b^	0.78 ± 0.03 ^b^	0.91 ± 0.05 ^a^
n-6 PUFA	38.57 ± 3.55 ^a^	34.28 ± 1.54 ^a,b^	32.11 ± 3.04 ^b^
18:3n-3	1.87 ± 0.20 ^b^	2.16 ± 0.07 ^a^	2.09 ± 0.29 ^a^
18:4n-3	0.24 ± 0.08	0.25 ± 0.03	0.20 ± 0.02
20:5n-3	0.40 ± 0.01^c^	0.68 ± 0.02 ^b^	0.89 ± 0.03 ^a^
22:6n-3	2.28 ± 0.32^c^	4.21 ± 0.12 ^b^	7.68 ± 0.28 ^a^
n-3 PUFA	5.85 ± 0.40^c^	8.02 ± 0.54 ^b^	11.63 ± 0.66 ^a^

Values are mean ± SEM (*n* = 3). It shows significant differences when the superscript letters of the values in the same line are completely different (*p* < 0.05). SFA: saturated fatty acid; MUFA: monounsaturated fatty acid; n-6 PUFA: omega-6 polyunsaturated fatty acids; n-3 PUFA: omega-3 polyunsaturated fatty acids.

**Table 5 foods-11-03600-t005:** The resulting muscle proximate compositions, edible quality, and textural properties.

	Experimental Diets		
	D1	D2	D3
Proximate composition (% wet matter)			
Moisture	76.20 ± 1.45	77.28 ± 2.34	77.02 ± 1.66
Crude protein	18.21 ± 0.23	17.89 ± 0.44	17.93 ± 0.55
Crude lipid	5.57 ± 0.47	4.94 ± 0.55	5.50 ± 0.63
Nutritional indices of FAs			
PUFA/SFA	1.93 ± 0.16	2.01 ± 0.13	2.07 ± 0.15
n-6/n-3 PUFA	6.57 ± 0.21 ^a^	4.26 ± 0.37 ^b^	2.87 ± 0.15 ^c^
FLQ	2.69 ± 0.18 ^c^	4.89 ± 0.11 ^b^	8.57 ± 0.27 ^a^
IA	0.22 ± 0.01	0.21 ± 0.02	0.21 ± 0.02
IT	0.38 ± 0.02 ^a^	0.32 ± 0.02 ^a^	0.26 ± 0.03 ^b^
HH	4.52 ± 0.10	4.64 ± 0.08	4.53 ± 0.05
Muscular edible quality			
pH	6.54 ± 0.02	6.53 ± 0.06	6.58 ± 0.03
COP/%	84.64 ± 2.43	88.32 ± 1.24	85.41 ± 1.60
WHC/%	6.14 ± 0.63 ^a^	6.84 ± 0.14 ^a^	7.61 ± 0.73 ^b^
Muscular textural properties			
Tenderness (gf)	11.04 ± 0.59	11.78 ± 0.37	11.12 ± 0.84
Hardness (gf)	297.67 ± 24.31	349.04 ± 18.29	398.12 ± 26.36
Adhesiveness (mJ)	4.09 ± 0.51	4.87 ± 0.51	4.94 ± 0.60
Springiness (mm)	0.47 ± 0.02	0.46 ± 0.02	0.44 ± 0.02
Chewiness (mJ)	73.39 ± 7.37	71.89 ± 8.66	72.04 ± 7.01
Gumminess (mJ)	108.75 ± 10.08 ^a^	147.35 ± 7.53 ^b^	144.08 ± 8.37 ^b^

Values are mean ± SEM (*n* = 3). Values of the same row with different superscript letters were significant differences (*p* < 0.05). PUFA/SFA: polyunsaturated fatty acids/saturated fatty acid; n-6/n-3 PUFA: omega-6/omega-3 polyunsaturated fatty acids; FLQ: fish lipid quality/flesh lipid quality; IA: index of atherogenicity; IT: index of thrombogenicity; HH: hypocholesterolemic/hypercholesterolemic ratio; COP: cooked meat percentage; WHC: water holding capacity. FLQ = 100 × (22:6 n-3 + 20:5 n-3)/ΣFA. IA = [12:0 + (4 × 14:0) + 16:0]/UFA. IT = (14:0 + 16:0 + 18:0)/[(0.5 × MUFA) + (0.5 × n-6 PUFA) + (3 × n-3 PUFA) + (n-3/n-6)]. HH = (18:1 + PUFA)/(12:0 + 14:0 + 16:0).

## Data Availability

Data is contained within the article or Appendix A.

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
