# Peer review of "Dietary DHA Enhanced the Textural Firmness of Common Carp (Cyprinus carpio L.) Fed Plant-Derived Diets through Restraining FoxO1 Pathways"

_foods, 2022, doi:10.3390/foods11223600_

Round 1

Reviewer 1 Report

The manuscript "Supplementation of DHA enhanced the textural firmness of common carp (Cyprinus carpio L.) fed plant-based diets through the FoxO1 pathways" is an interesting topic. However, I would suggest making some corrections (attached file) to improve your submission for Foods journal.

Author Response

Many thanks for the comments from the reviewers, which enable us to revise and improve our manuscript (foods-1972570). According to the reviewer’s suggestion, the point-by-point responses to each comment are provided below.

Title:The title is appropriate because it summarizes the main result, however, the use of the words "textural firmness" is more appropriate for postharvest products (e.g., fruits and vegetables); it is suggested to use the words "textural muscle" to indicate that it is about the muscle (the change is suggested in the rest of the text).

Response: The "textural firmness" is often used to describe the postharvest products of fruits and vegetables. While, the “textural muscle” or “muscle texture” is a comprehensive concept, which includes tenderness, hardness, adhesiveness, springiness, chewiness, gumminess, cohesiveness, resilience. Among texture attributes, hardness also termed as firmness, an essential evaluating parameter of fish freshness is closely associated with the human visible acceptability of fish products (Cheng et al., 2014, Food Research International, 56, 190-198; Yu et al., 2020,Food chemistry, 325, 126906.). Additionally, the muscle hardness is closely linked to muscle gumminess and chewiness.

Thus, we suggested that the textural firmness is suitable for keeping in the MS.

Cheng, J. H., Qu, J. H., Sun, D. W., et a;. (2014). Visible/near-infrared hyperspectral imaging prediction of textural firmness of grass carp (Ctenopharyngodon idella) as affected by frozen storage. Food Research International, 56, 190-198.

Yu, E., Fu, B., Wang, G., et al. (2020). Proteomic and metabolomic basis for improved textural quality in crisp grass carp (Ctenopharyngodon idellus C. et V) fed with a natural dietary pro-oxidant. Food chemistry, 325, 126906.

Abstract: It is necessary to start with a "problematic" to put the reader in context, about the reason for the study.

Response: Thanks for your suggestion. “Omega-3 fatty acid have a positive effect on the muscle textural firmness of fish, while the intrinsic mechanism is poorly understood” was inserted in the revised abstract.

Introduction: The information is appropriate; however, it could be organized better.

Response: Revision has been made according to your suggestion. Details please see the revised MS..

Material and method: The methodology has adequate information, is developed, and organized correctly. Lines 76-77. Indicate that the values 8:30, 13:00, 17:30, and 18:30 are hours.

Response: Yes, the values of 8:30, 13:00, 17:30, and 18:30 indicate the time of feeding and changing water. The “am” and “pm” were inserted behind the time values.

Results: The section is adequate, and it has an organized development; however, it is important to correct some details: Table 4. The meaning of the letters in SFA, MUFA, n-6 PUFA and n-3 PUFA is unclear. Table 5. The meaning of the letters in PUFA/SFA, n-6/n-3 PUFA, FLQ, IA, IT, HH, COP, and WHC is unclear. Figure 1. The figure is composed of too many images that are not sufficiently explained in the main text. Figure 2. The figure is not sufficiently explained in the main text. Figure 3. The content of the figure is illegible. The figure is not sufficiently explained in the main text. Figure 4. The figure is not sufficiently explained in the main text. Figure 5. The figure is not sufficiently explained in the main text.

Response: Revision has been made according to your suggestion. Details please see the revised MS.

The size of original Figure 3 is too big, the content of the figure 3 is illegible when the figure 3 was minified and inserted in the MS. Accordingly, the original Figure 3 was revised as Figure S4, and uploaded as a Supplemental material.

Discussion: The section is adequate, has an organized development, appropriate comparisons, and sufficient analysis; however, it is important to correct some details:

Response: Revision has been made according to your suggestion. Details please see the revised Discussion.

Conclusions: It is not appropriate to start the paragraph with the words "in summary" because it is not a summary, it is a conclusion. It is suggested to change "textural firmness to"textural muscle"(as in the title)

Response:Thanks for your suggestion. "in summary" has been revised as “in conclusion”. As we explained the first question of Reviewer 2, we suggested that the textural firmness is suitable for keeping in the MS.

References: The references are adequate, however only 50% are current (last 5 years).

Response: Recent references have been cited according to your suggestion. Details please see the revised References.

Reviewer 2 Report

This manuscript aimed to investigate the effect of DHA dietary inclusion on the muscle texture of common carp and its potential regulatory mechanism. In general, the manuscript is well written, the organization and the structure of the manuscript are quite satisfactory. The experimental design is properly organized and set, while the presented results are explained and discussed in concise and comprehensive way. The results contribute to the better understanding of the potential mechanism of DHA dietary inclusion in common carp diet on the improvement in muscle textural firmness. More specific comments are outlined below:

The authors mentioned that diets were air-dried at room temperature and stored, whereas there was no any information on diet production. Please provide a few details about it.

Please provide information on temperature regimes, column and detector (FID/MS) employed for the fatty acid analysis.

Page 6, lines 213-214: The authors stated that MUFA increased with the increasing dietary DHA level which was not the case. MUFA as well as n-6 PUFA decreased with the increasing DHA level.

Page 7, lines 232-233: The authors reported that muscle lipid content was significantly affected by dietary DHA. According to Table 5, there were no significant differences between groups for crude lipid content. Please, check it. The same comment stands for hardness.

Figures 1F, 1G, 1H, 1I were not explained in figure caption. Please provide appropriate explanation.

Author Response

Many thanks for the comments from the reviewers, which enable us to revise and improve our manuscript (foods-1972570). According to the reviewer’s suggestion, the point-by-point responses to each comment are provided below.

The authors mentioned that diets were air-dried at room temperature and stored, whereas there was no any information on diet production. Please provide a few details about it.

Response: Thanks for your suggestion. Diets were prepared and stored as described in our previous study (Li et al., 2020, Aquaculture, 516, 734632). Briefly, the powdered ingredients were first mixed thoroughly by hand. Next, the dietary powder was mixed and homogenized with oil and distilled water by hand. Pellets (Φ2 mm) were made using an automatic pellet-making machine (SLC-45, Fishery Machinery and Instrument Research Institute, China). Diets were air-dried to approximately 10% moisture, then sealed in vacuum-packed bags and stored at −20 ℃ until use.

Accordingly, “The preparation of diets were carried out as we previously reported in detail” was inserted in the revised MS.

 Please provide information on temperature regimes, column and detector (FID/MS) employed for the fatty acid analysis.

Response: Thanks for your suggestion. The detailed GC parameters were employed following the previous study (Xie et al. 2022, Journal of Agricultural and Food Chemistry, 70(8), 2701-2711). Briefly, FAME samples were applied using on-column (HP-88, 100 m × 0.25 mm × 0.2 μm, USA, Agilent) injection with the oven temperature programmed to rise from 80 to 250 ℃ at 40 ℃ min−1, and individual fatty acids were identified by comparison against commercial standards (Sigma) and quantified with CLASS-GC2010-plus workstation (Shimadzu). Details please see the revised MS. 

Page 6, lines 213-214: The authors stated that MUFA increased with the increasing dietary DHA level which was not the case. MUFA as well as n-6 PUFA decreased with the increasing DHA level.

Response: Sorry, it is our carelessness. The n-3 PUFA contents increased gradually with the increasing dietary DHA levels, whereas n-6 PUFA and MUFA displayed an opposite trend (P < 0.05). Revision has been made, details please see the revised MS. 

Page 7, lines 232-233: The authors reported that muscle lipid content was significantly affected by dietary DHA. According to Table 5, there were no significant differences between groups for crude lipid content. Please, check it. The same comment stands for hardness.

Response: Thanks for your reminding, there were inaccurate portrayal of muscle lipid contents and hardness. The muscle crude lipid contents shared comparable values among the three dietary treatments (P > 0.05). Although the muscular hardness showed no statistical difference among the three groups, the numerical value of those indexes in the D2 and D3 groups was better than that in D1 group. Revision has been made, details please see the revised MS. 

Figures 1F, 1G, 1H, 1I were not explained in figure caption. Please provide appropriate explanation.

Response: Revision has been made according to your suggestion. Details please see the caption of revised Figure 1. 

Round 2

Reviewer 1 Report

The manuscript is correct, only one detail needs to be corrected:

Line 84-85. If the authors want to use the terms "am" and "pm" they must organize the hours in a 12-hour clock, if the authors are going to use "h" they must use a 24-hour clock. In other words, their hours would be: 8:30 a.m., 1:00 p.m., 5:30 p.m. and 6:30 p.m. or 8:30 a.m., 1:00 p.m., 5:30 p.m. and 6:30 p.m.

Author Response

Your comments, which enabled us to revise and improve our manuscript (foods-1972570), are greatly appreciated. According to your suggestion, the reply is provided below.

Line 84-85. If the authors want to use the terms "am" and "pm" they must organize the hours in a 12-hour clock, if the authors are going to use "h" they must use a 24-hour clock. In other words, their hours would be: 8:30 a.m., 1:00 p.m., 5:30 p.m. and 6:30 p.m. or 8:30 a.m., 1:00 p.m., 5:30 p.m. and 6:30 p.m.

Response: Thanks for your suggestion. "13:00 pm, 17:30 pm, and 18:30 pm" have been revised as “1:00 pm, 5:30 pm, 6:30 pm”. Details please see the revised MS.
